# Noise-Robust-Based Clock Parameter Estimation and Low-Overhead Time Synchronization in Time-Sensitive Industrial Internet of Things

**DOI:** 10.3390/e27090927

**Published:** 2025-09-03

**Authors:** Long Tang, Fangyan Li, Zichao Yu, Haiyong Zeng

**Affiliations:** 1Guangxi Key Laboratory of Braininspired Computing and Intelligent Chips, School of Electronic and Information Engineering, Guangxi Normal University, Guilin 541001, China; 17823537490@163.com (L.T.); lifangyan@mailbox.gxnu.edu.cn (F.L.); 2Department of Electronic Engineering and Information Science, University of Science and Technology of China, Hefei 230022, China; zichaoyu@mail.ustc.edu.cn

**Keywords:** time synchronization, Maximum Likelihood Estimation, parameter estimation, Industrial Internet of Things

## Abstract

Time synchronization is critical for task-oriented and time-sensitive Industrial Internet of Things (IIoT) systems. Nevertheless, achieving high-precision synchronization with low communication overhead remains a key challenge due to the constrained resources of IIoT devices. In this paper, we propose a single-timestamp time synchronization scheme that significantly reduces communication overhead by utilizing the mechanism of AP to periodically collect sensor device data. The reduced communication overhead alleviates network congestion, which is essential for achieving low end-to-end latency in synchronized IIoT networks. Furthermore, to mitigate the impact of random delay noise on clock parameter estimation, we propose a noise-robust-based Maximum Likelihood Estimation (NR-MLE) algorithm that jointly optimizes synchronization accuracy and resilience to random delays. Specifically, we decompose the collected timestamp matrix into two low-rank matrices and use gradient descent to minimize reconstruction error and regularization, approximating the true signal and removing noise. The denoised timestamp matrix is then used to jointly estimate clock skew and offset via MLE, with the corresponding Cramér–Rao Lower Bounds (CRLBs) being derived. The simulation results demonstrate that the NR-MLE algorithm achieves a higher clock parameter estimation accuracy than conventional MLE and exhibits strong robustness against increasing noise levels.

## 1. Introduction

In recent years, significant research and practical applications have focused on the task-oriented Industrial Internet of Things (IIoT), which integrates wireless communication technologies, mobile computing platforms, and distributed sensor networks [1,2]. Maintaining precise temporal alignment emerges as a fundamental necessity for time-critical IIoT applications, becoming increasingly crucial in modern industrial automation systems to optimize operational performance, enable deterministic low-latency communication, and minimize power usage [3,4]. Generally, individual industrial equipment establishes its internal clock by monitoring oscillation cycles from its integrated crystal resonator. Nevertheless, production tolerances and changing operational conditions can lead to frequency inaccuracies and progressive timing shifts, causing clock discrepancies among networked devices that may induce unpredictable delays in distributed control loops. Therefore, periodic time data exchange becomes essential for maintaining synchronized operations across the network while preserving bounded end-to-end latency. Additionally, power efficiency represents a major design consideration in IIoT deployments since it significantly determines system sustainability [5]. Thus, synchronization algorithms should emphasize low-power operation and reduced processing demands without compromising the synchronization accuracy that underpins real-time performance [2].

The straightforward approach to improving the energy efficiency of time synchronization is to reduce the number of required timestamp packet transmissions [6]. Capitalizing on the broadcast nature of wireless networks, researchers have proposed one-way message dissemination, which consumes fewer transmissions than traditional two-way message exchange [7]. Prior works [8,9] introduced a one-way time synchronization scheme for energy-efficient LoRa networks and a microsecond-level synchronization method based on low-layer timestamping, both of which improve time synchronization energy efficiency. Further enhancing efficiency, receiver-only synchronization (ROS) enables passive nodes to synchronize by overhearing communications between active node pairs [10]. Similarly, the reverse asymmetric time synchronization framework [11] was introduced to minimize energy consumption. However, these methods either rely on dedicated synchronization packets or require timestamp pairs, which may still impose energy or computational overheads.

One of the primary challenges pertaining to time synchronization in Industrial Internet of Things systems is random delay modeling [12]. In the absence of such delays during timestamp packet exchange between nodes, the relative clock parameters can be easily estimated. The comprehensive transmission delays between the target node and the reference node can be segregated into two types—fixed delay and random delay [11,13]. Fixed delays comprise transmission, reception, and propagation delays, and are significantly influenced by factors such as the speed of the radio, packet size, and link distance, with the delays varying depending on the distances between unsynchronized nodes and APs. Conversely, random delays, encompassing access delays, send delays, receive delays, interrupt, and encoding/decoding delays, are mainly affected by access delay, which is a principal source of error. The magnitude of the access delay is primarily dictated by the specific channel contention characteristics. Extensive research has addressed time synchronization under random network delays in timestamp packet transmission. Generally, random delays can be modeled as a random variable (RV) following Gaussian [14,15], exponential [16,17], or Weibull [18] distributions. The authos of [19] demonstrated that the synchronization process (packet transmission and receiving) can be generally regarded as a number of independent procedures; according to the central limit theorem, random delays can be modeled as a Gaussian distributed variable with more than 99.8% confidence.

Multiple studies have investigated the joint estimation of clock parameters using MLE under varying delay assumptions [14,15,16,20,21,22]. However, for fixed observation samples, the Cramér–Rao lower bound of the MLE, governed by the Fisher information and noise level, imposes a fundamental limit on estimation accuracy. This accuracy deteriorates markedly as the random delay noise intensifies [12,20]. As stated in [20], increasing the delay variance from 0.1 to 0.5 degrades the clock skew estimation performance by a factor of 4.9. Further increasing the variance to 1 results in a performance degradation of 9.8 times. Therefore, in scenarios with significant random delay noise, relying solely on MLE for time synchronization faces clock parameter estimation accuracy limitations, highlighting the need for developing noise-resistant algorithms.

In light of the aforementioned discussion, in this paper, we propose a noise-robust-based Maximum Likelihood Estimation (NR-MLE) algorithm that integrates advanced matrix decomposition techniques with optimized gradient descent methods, as well as a single-timestamp time synchronization scheme for IIoT. Our contributions are summarized as follows:We propose the noise-robust-based Maximum Likelihood Estimation (NR-MLE) algorithm to address noise-induced performance degradation in time synchronization parameter estimation. Specifically, we decompose the collected timestamp matrix into the product of two low-rank matrices. By employing gradient descent to minimize reconstruction errors and regularization terms, the decomposition approximates the low-rank structure of the true signal, thereby separating noise and yielding a denoised timestamp matrix. This enhances the precision of clock parameter estimation. Furthermore, the denoised timestamps enable the joint estimation of clock skew and offset via MLE. Moreover, we derive the corresponding Cramér–Rao Lower Bounds (CRLBs) to validate the estimation accuracy. The simulation results demonstrate that the proposed NR-MLE algorithm achieves a significant improvement in parameter estimation performance compared to the existing MLE algorithm.We propose a single-timestamp time synchronization scheme for IIoT, leveraging one-way message dissemination in sensor networks. The scheme utilizes the existing mechanism, where an access point (AP) periodically collects data from sensor nodes. When synchronization is triggered, the node sends its current timestamp to the AP during the first round. In subsequent rounds, the node omits timestamp transmissions, while the AP records the arrival time of the node’s data packets. Based on the collected timestamps and arrival times, the AP estimates the clock skew and offset between itself and the node, significantly reducing communication overhead.

The remainder of this paper is organized as follows: Section 2 introduces the system model; Section 3 introduces the proposed NR-MLE algorithm and the estimation of clock parameters; Section 4 provides simulation results and analysis; Section 5 concludes the paper and discusses future research directions.

## 2. System Model

### 2.1. Clock Model

Time synchronization is a fundamental and crucial technology in the task-oriented and time-sensitive IIoT. In automated production systems, industrial mobile robots work together on production lines by receiving and executing commands to achieve unmanned manufacturing. To enable effective collaboration, all participating devices must share a consistent time reference, creating the need for high-precision clock synchronization, which is also essential for achieving deterministic low-latency communication in time-critical applications.

Although each industrial robot maintains its own internal clock, various uncontrollable factors—such as temperature fluctuations, pressure variations, and oscillator aging—can cause timing discrepancies that accumulate over time [23]. Similarly, in Industrial Internet of Things systems, every node and AP relies on its internal clock. Under ideal conditions, a node’s local time could be expressed as C(t), where *t* represents the standard time. However, imperfections in clock oscillators (e.g., aging or thermal effects) introduce frequency offsets (or clock skew), causing local clocks to diverge from the reference time [24]. Typically, a node’s local time C(t) can be modeled as follows:(1)C(t)=βt+α,
where β and α represent clock skew and clock offset, respectively. Clock skew is the frequency deviation of the target node’s clock relative to the reference clock source, while clock offset is the phase deviation of the target node’s clock relative to the reference clock source. As shown in Figure 1, the solid line in the figure represents the clock model of target node *S*, the dashed line represents the clock model of the AP, the intercept of the curve with the y-axis represents the clock offset, and the slope of the curve represents clock skew.

Clearly, when β = 1 and α = 0, the target node remains perfectly aligned with the reference clock. It is important to highlight that while clock offset directly impacts synchronization performance, clock skew is equally crucial as it causes gradual desynchronization. Thus, periodic skew estimation and compensation are vital for sustaining high-precision synchronization over extended durations.

### 2.2. Delay Model

A key challenge in achieving time synchronization for Industrial Internet of Things systems lies in modeling random delays. The total transmission delays between reference and target nodes can be categorized into two components—fixed delays and random delays. Generally, the random delays can be modeled as a random variable (RV) following Gaussian, exponential, or Weibull distributions. It has been demonstrated in [19] that the synchronization process (packet transmission and receiving) can generally be regarded as a number of independent procedures; according to the central limit theorem, the random delay can be modeled as a Gaussian distributed variable with more than 99.8% confidence. This assumption aligns with prior works in wireless network time synchronization [14,15,20,21], where Gaussian distribution serves as a universal baseline for theoretical analysis and algorithm design. The central limit theorem’s applicability to aggregated delays (transmission, reception, queuing, etc.) ensures the model’s robustness for general IIoT scenarios. Based on this fact, we choose a Gaussian random delay model in this paper.

### 2.3. Proposed Single-Timestamp Time Synchronization Model

As illustrated in Figure 2, we propose a single-timestamp time synchronization scheme for IIoT, where the AP serves as the reference time source and node *S* is the device to be synchronized. The scheme leverages the existing periodic data collection mechanism with a fixed transmission period δ. The synchronization proceeds as follows:First Round (i=1):-Node *S* sends a single synchronization timestamp $T_{1}^{S}$ to the AP.-The AP records its local reception time as $T_{2,1}^{R}$.Subsequent Rounds (i=2,3,…,N):-Node *S* transmits only sensor data (no additional timestamps) to the AP every δ time units.-The AP records each reception time $T_{2,i}^{R}$.Final Steps-After *N* rounds of synchronization, the AP obtains a set of timestamp information ${\left\{T_{1}^{S},T_{2,i}^{R}\right\}}_{i=1}^{N}$, which is used to estimate the clock skew $\hat{\beta }$ and offset $\hat{\alpha }$ relative to node *S*. Then, they are sent to node *S* to adjust their own clock parameters, ultimately achieving time synchronization.

**Figure 2 entropy-27-00927-f002:**
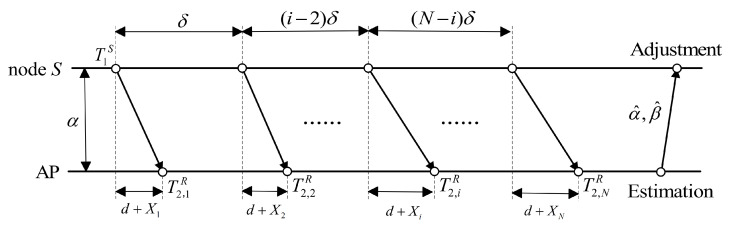
Proposed single-timestamp time synchronization scheme based on one-way message dissemination for IIoT, where node *S* records and transmits one timestamp to the AP in the first synchronization round, whereas in the subsequent rounds, node S sends the data to be collected by the AP at a fixed period δ.

According to the clock model of Equation (Equation 1) and the collected timestamp information ${\left\{T_{1}^{S},T_{2,i}^{R}\right\}}_{i=1}^{N}$, the above procedure can be modeled as follows:(2)$$T_{2,i}^{R}=\frac{\left(T_{1}^{S}+\left(i-1\right)\delta \right)}{\beta }-\frac{\alpha }{\beta }+d+X_{i},\forall i=1,2,...,N,$$
where β and α represent the clock skew and clock offset of AP with respect to node *S*, respectively; *d* represents the fixed packet delays, which are unknown but remain constant; and $X_{i}$ depicts the random delays during the *i*-th round. Based on the reasons explained in Section 2.2, $X_{i}$ is assumed to independently obey a Gaussian distribution with zero mean and variance $\sigma^{2}$, i.e., $X_{i}\overset{iid}{∼}N\left(0,\sigma^{2}\right)$.

Based on Equation (Equation 2), Maximum Likelihood Estimation can be executed utilizing a set of matrix timestamps. Assuming that, after *N* synchronization events, a linear timestamp model can be established as follows:(3)$$\underset{T^{R}}{\underset{︸}{\left[\begin{array}{c} T_{2,1}^{R}\\ \vdots \\ T_{2,i}^{R}\\ \vdots \\ T_{2,N}^{R} \end{array}\right]}}=\underset{H}{\underset{︸}{\left[\begin{array}{ccc} T_{1}^{S} & -1 & 1\\ \vdots & \vdots & \vdots \\ T_{1}^{S}+\left(i-1\right)\delta & -1 & 1\\ \vdots & \vdots & \vdots \\ T_{1}^{S}+\left(N-1\right)\delta & -1 & 1 \end{array}\right]}}\underset{\theta }{\underset{︸}{\left[\begin{array}{c} \theta_{1}\\ \theta_{2}\\ \theta_{3} \end{array}\right]}}+\underset{X}{\underset{︸}{\left[\begin{array}{c} X_{1}\\ \vdots \\ X_{i}\\ \vdots \\ X_{N} \end{array}\right]}}$$
which includes the following parameter transformation:(4)$$\theta_{1}=\frac{1}{\beta },\theta_{2}=\frac{\alpha }{\beta },\theta_{3}=d.$$

In this model, $T^{R}$ denotes the observation vector, H represents the observation matrix, θ encapsulates the parameters to be estimated, and X is indicative of the noise vector.

## 3. Noise-Robust-Based Clock Parameter Estimation

In this section, we address the critical challenge of random delay noise in timestamp-based clock parameter estimation, which significantly deteriorates the accuracy of Maximum Likelihood Estimation. To overcome this limitation, we propose a novel noise-robust-based Maximum Likelihood Estimation (NR-MLE) algorithm that integrates advanced matrix decomposition techniques with optimized gradient descent methods. The proposed solution operates through a two-phase approach. First, it systematically preprocesses the noisy timestamp matrix by decomposing and reconstructing its core components to isolate and mitigate random delay effects; then, it applies refined MLE estimation on the denoised data to achieve enhanced parameter estimation accuracy. This comprehensive framework not only improves robustness against random delays but also maintains the statistical efficiency of traditional MLE.

### 3.1. Problem Formulation

Based on the collected timestamp information ${\left\{T_{1}^{S},T_{2,i}^{R}\right\}}_{i=1}^{N}$ and Equation (Equation 3), we define the matrices $T^{S}\in R^{N\times 1}$ and $T^{R}\in R^{N\times 1}$ as follows:(5)$$\begin{array}{c} \begin{array}{cc} T^{S} & ={\left[\begin{array}{ccc} T_{1}^{S} & \ldots & T_{1}^{S}+ \end{array}\left(N-1\right)\delta \right]}^{T},\\ T^{R} & ={\left[\begin{array}{ccc} T_{2,1}^{R} & \ldots & T_{2,N}^{R} \end{array}\right]}^{T}. \end{array} \end{array}$$

According to Equation (Equation 3) and the proposed single-timestamp time synchronization mechanism in Figure 2, it can be observed that random delay noise is reflected on $T^{R}$. Consequently, the timestamp matrix $T^{SR}\in R^{N\times 2}$ containing random delay noise can be represented as follows:(6)$$T^{SR}=\left[\begin{array}{cc} T^{S} & T^{R} \end{array}\right]=T_{ideal}^{SR}+W,$$
where

$T_{\mathrm{ideal}}^{SR}=\left[\begin{array}{cc} T^{S} & H\theta \end{array}\right]$ is the ideal rank-1 matrix, which has not been contaminated by random delay noise.$W=\left[\begin{array}{cc} 0_{N\times 1} & X \end{array}\right]$ is the noise matrix with $X_{i}\overset{\mathrm{iid}}{∼}N\left(0,\sigma^{2}\right)$.

Our goal is to obtain the ideal timestamp matrix $T_{ideal}^{SR}$ from the collected timestamp matrix $T^{SR}$; therefore, we need to search for factor matrices U∈$R^{2\times p}$ and V∈$R^{N\times p}$ from $T^{SR}$, where p≪min(2,N), which satisfies the following:(7)$$T_{\mathrm{ideal}}^{SR}=UV^{T}.$$

### 3.2. Optimization Objectives

#### 3.2.1. Objective Function Design and Motivation

The core objective of our matrix factorization approach is to recover a clean low-rank matrix from noisy observations while maintaining the essential structure of the data. This is achieved by minimizing the composite objective function that balances reconstruction accuracy against noise suppression. The Frobenius norm term $T^{SR}-U{V^{T}}_{F}^{2}$ ensures faithful signal reconstruction, while the regularization terms $\lambda (U_{F}^{2}+V_{F}^{2})$ enforce low-rank constraints to suppress noise. The parameter λ controls this trade-off, with larger values emphasizing noise reduction and smaller values preserving signal details. This dual optimization approach effectively separates the underlying temporal structure from random delay noise while maintaining computational efficiency. Consequently, we seek to solve the following optimization problem:    (8)$$\min_{U,V}f\left(U,V\right)=∥T^{SR}-{\mathrm{UV}}^{T}{∥}_{F}^{2}+{\lambda (∥U∥}_{F}^{2}+{∥V∥}_{F}^{2})$$
where $T^{SR}\in R^{2\times N}$ is the noisy input matrix; $U\in R^{2\times p}$ and $V\in R^{N\times p}$ are the factor matrices; ${∥\cdot ∥}_{F}$ is the Frobenius norm; λ is the regularization coefficient; and p≪min(2,N) is the target rank that captures the essential signal dimensions.

This is achieved through a carefully designed objective function that balances the following two competing goals:Faithful Data Reconstruction: The first term $∥T^{SR}-{\mathrm{UV}}^{T}{∥}_{F}^{2}$ ensures the factorized product ${\mathrm{UV}}^{T}$ closely approximates the observed data. The Frobenius norm is particularly suitable for this purpose [25,26] due to the following reasons:-It provides a smooth, differentiable measure of the total reconstruction error;-It corresponds to the Maximum Likelihood Estimator under Gaussian noise assumptions;-Its quadratic nature leads to computationally efficient optimization;-It equally weights all matrix elements, making it appropriate when errors are uniformly distributed across observations.Effective Noise Suppression: The regularization terms $\lambda (∥U{∥}_{F}^{2}+∥V{∥}_{F}^{2})$ play several crucial roles in denoising [27,28], as follows:-They prevent overfitting to noise by penalizing large values in the factor matrices;-They impose implicit rank constraints by shrinking smaller singular values more aggressively (as can be seen through the lens of singular value shrinkage);-They improve numerical stability during optimization.

Note that the parameter λ controls the trade-off between these two objectives [29]. Larger values of λ lead to stronger denoising but may oversmooth genuine signal components, while smaller values preserve more details but may retain noise. This trade-off can be interpreted through the bias-variance decomposition—increasing λ reduces variance (noise) at the cost of increased bias (signal distortion). The optimal λ can be selected through cross-validation or theoretical considerations like the discrepancy principle when noise statistics are known.

#### 3.2.2. Gradient Computation

The optimization proceeds via alternating gradient descent, where the gradients have intuitive interpretations, as follows:

For matrix U:(9)$$\begin{array}{cc} \nabla_{U}f & =-2(T^{SR}-{\mathrm{UV}}^{T})V+2\lambda U \end{array}$$(10)$$\begin{array}{cc} \; & =-2(\mathrm{Reconstruction}\;\mathrm{error})V+2\lambda U \end{array}$$

The first term adjusts U to reduce the reconstruction error, weighted by the current V matrix. The second term pulls U toward zero, implementing the denoising effect.

Similarly, for V:(11)$$\begin{array}{cc} \nabla_{V}f & =-2{\left(T^{SR}-{\mathrm{UV}}^{T}\right)}^{T}U+2\lambda V \end{array}$$(12)$$\begin{array}{cc} \; & =-2{\left(\mathrm{Reconstruction}\;\mathrm{error}\right)}^{T}U+2\lambda V \end{array}$$

#### 3.2.3. Convergence Analysis

The algorithm’s convergence is monitored through following metric:Primary Stopping Criterion: The relative change in normalized reconstruction error, as follows:(13)$$\frac{|\epsilon_{t}-\epsilon_{t-1}|}{\epsilon_{t-1}}<\tau$$
where $\epsilon_{t}=∥T^{SR}-U^{\left(t+1\right)}V^{(t+1)T}{∥}_{F}/{∥T^{SR}∥}_{F}$; *t* denotes the *t*-th iteration; and τ is typically set between $10^{-4}$ and $10^{-6}$ for high-precision applications.Convergence Guarantees-The objective function is guaranteed to decrease monotonically;-Convergence to a local minimum is assured due to the convexity of subproblems;-The alternating minimization strategy avoids poor local minima, which are common in joint optimization.

The denoising performance can be qualitatively assessed by examining the residual matrix $T^{SR}-{\mathrm{UV}}^{T}$, which should appear as random noise without discernible structure when the factorization is successful.

### 3.3. Joint Clock Parameter Estimation

After satisfying the convergence condition, the denoised timestamp matrix $T_{D}^{SR}$ can be obtained and approaches the ideal timestamp matrix $T_{\mathrm{ideal}}^{SR}$, in order to further obtain the denoised timestamp vector $T_{D}^{R}$. Consequently, we establish the denoised linear model as follows:(14)$$T_{D}^{R}=H\theta +\tilde{X},$$
where $\tilde{X}$ is the residual noise matrix. We assume that the residual noise after denoising still follows a Gaussian distribution, with a mean of zeros and unknown variance, i.e., $X_{i}\overset{iid}{∼}N\left(0,s^{2}\right)$. Then, the logarithm likelihood function is given as in [30,31], as follows:(15)$$\ln p\left(T_{D}^{R};\theta \right)=\frac{N}{2}\ln\frac{1}{2\pi s^{2}}-\frac{{∥T_{D}^{R}-H\theta ∥}^{2}}{2s^{2}}.$$

By computing the partial derivative of the vector parameter θ, with respect to the parameters to be estimated, and setting it to zero, the Maximum Likelihood Estimate of $\hat{\theta }$ is as follows:(16)$$\hat{\theta }={\left({\left(H\right)}^{T}H\right)}^{-1}{\left(H\right)}^{T}T_{D}^{R}.$$

The matrix ${\left(H\right)}^{T}H$ is invertible since the rank of the matrix is equal to its dimension (full rank) [30,31]. Thus, the estimated clock skew $\hat{\beta }$ and clock offset $\hat{\alpha }$ based on Equation (16) are expressed as follows:(17)$$\hat{\beta }=\frac{1}{{\hat{\theta }}_{1}},$$(18)$$\hat{\alpha }=\frac{{\hat{\theta }}_{2}}{{\hat{\theta }}_{1}}.$$

The proposed NR-MLE algorithm is summarized in Algorithm 1. In the algorithm, $\eta_{t}$ is the learning rate at the *t*-th iteration, controlling the step size for updating the parameters U and V in gradient descent. $I_{\max}$ denotes the maximum number of iterations, a predefined stopping criterion if convergence is not reached earlier.
**Algorithm 1** Proposed NR-MLE algorithm.**Require:** Reducing the noise of the collected timestamp matrix $T^{SR}$ and estimating clock parameters β and α.
1:**Input**: Noisy matrix $T^{SR}$, target rank *p*, regularization λ.2:**Initialize**: Random $U^{\left(0\right)}\in R^{2\times p}$, $V^{\left(0\right)}\in R^{N\times p}$3:**for** t=0 to $I_{max}-1$ **do**4:    Compute gradient: $G_{U}=-2\left(T^{SR}-U^{\left(t\right)}V^{(t)T}\right)V^{\left(t\right)}+2\lambda U^{\left(t\right)}$5:    Update U: $U^{\left(t+1\right)}=U^{\left(t\right)}-\eta_{t}G_{U}$6:    Compute gradient: $G_{V}=-2{\left(T^{SR}-U^{\left(t+1\right)}V^{(t)T}\right)}^{T}U^{\left(t+1\right)}+2\lambda V^{\left(t\right)}$7:    Update V: $V^{\left(t+1\right)}=V^{\left(t\right)}-\eta_{t}G_{V}$8:    Compute normalized error: $\epsilon_{t}=∥T^{SR}-U^{\left(t+1\right)}V^{(t+1)T}{∥}_{F}/{∥T^{SR}∥}_{F}$9:    **if** $|\epsilon_{t}-\epsilon_{t-1}|<\tau \cdot \epsilon_{t-1}$ **then**10:        **break**11:    **end if**12:**end for**13:**Output**: Denoised matrix $T_{D}^{SR}$ = $U^{\left(t\right)}V^{(t)T}$14:Construct a denoised linear model: $T_{D}^{R}=H\theta +\tilde{X}$15:Equation (Equation 16) and can be used to estimate the clock offset $\hat{\alpha }$ and clock skew $\hat{\beta }$ of node *S* relative to the reference clock.


### 3.4. CRLB for Clock Offset and Skew

The closed-form expression for the CRLB pertaining to the estimation vector is ascertainable by inverting the 2 × 2 Fisher information matrix I(θ) [30,31]. This involves calculating the second-order partial derivatives of Equation (Equation 15), which concern the parameters α and β as follows:(19)$$\frac{\partial^{2}\ln p\left(T_{D}^{R};\theta \right)}{\partial \alpha^{2}}=-N\frac{1}{s^{2}\beta^{2}},$$(20)$$\begin{array}{c} \frac{\partial^{2}\ln p\left(T_{D}^{R};\theta \right)}{\partial \alpha \partial \beta }=\frac{\partial^{2}\ln p\left(T_{D}^{R};\theta \right)}{\partial \beta \partial \alpha }\\ =\frac{1}{s^{2}}\sum_{i=1}^{N}\left(\frac{T_{2,i}^{R}-d}{\beta^{2}}-\frac{\left(T_{1}^{S}+\left(i-1\right)\delta )-\alpha \right)}{\beta^{3}}\right), \end{array}$$(21)$$\begin{array}{c} \frac{\partial^{2}\ln p\left(T_{D}^{R};\theta \right)}{\partial \beta^{2}}=-\frac{1}{s^{2}}\sum_{i=1}^{N}\left(\frac{3{\left(\left(T_{1}^{S}+\left(i-1\right)\delta \right)-\alpha \right)}^{2}}{\beta^{4}}\right.\\ +\left.\frac{2\left(\alpha -\left(T_{1}^{S}+\left(i-1\right)\delta \right)\right)\left(T_{2,i}^{R}-d\right)}{\beta^{3}}\right). \end{array}$$

Taking Equations (19)–(21) to the negative expectations, we can obtain the Fisher information matrix FIM(θ) by inverting the Fisher information matrix FIM(θ); the closed-form CRLB for the clock skew and offset can take the following expressions, respectively:(22)$$Var\left(\hat{\beta }\right)\;⩾\;\frac{s^{2}N}{\beta^{2}\left|\mathrm{FIM}(\theta )\right|},$$(23)$$\begin{array}{c} Var\left(\hat{\alpha }\right)\;⩾\;\frac{s^{2}}{\left|\mathrm{FIM}(\theta )\right|}\;\sum_{i=1}^{N}\;\frac{2\left(\alpha \;-\;\left(T_{1}^{S}\;+\;\left(i\;-\;1\right)\delta \right)\right)\left(T_{2,i}^{R}\;-\;d\right)}{\beta^{3}}\\ +\frac{3{\left(\left(T_{1}^{S}+\left(i-1\right)\delta \right)-\alpha \right)}^{2}}{\beta^{4}}. \end{array}$$

## 4. Simulation Results and Analyses

### 4.1. Parameter Settings

The simulations were conducted using MATLAB R2023a on a Windows 11 system with Intel Core i7-12700H processor and 32 GB RAM, implementing a realistic IIoT environment simulation where the reference clock was modeled as ideal (β=1, α=0) while node clocks were simulated with skew β∈[0.99,1.01] (representing ±100 ppm oscillator variations), offset α∈[−0.02,0.02] ms, and μs-resolution timekeeping, incorporating fixed delays d∈[0,0.2] ms (constant per simulation run) and random delays $X_{i}∼N\left(0,\sigma^{2}\right)$ with $\sigma^{2}\in \left[0.1,1\right]$ being generated using MATLAB’s randn function. The synchronization protocol was executed with fixed δ=1 ms intervals and 1μs timestamp precision under packet-loss-free conditions. The complete parameters are summarized in Table 1 and each data point represents averages of max=10,000 independent Monte Carlo runs for statistical significance. The performance was evaluated using MSE between estimated parameters and ground-truth values.

### 4.2. Simulation Results

Figure 3 and Figure 4 compare the MSEs of clock skew and offset of the proposed single-timestamp time synchronization scheme and NR-MLE algorithm with that of the conventional timetamps scheme based on one-way message dissemination [12,32] and two-way message exchange [12] with respect to the number of synchronization rounds *N* when the variance of random delay $\sigma^{2}$ = 1. As the synchronization rounds *N* increase, a significant improvement in the estimation accuracy of both clock skew and offset is observed. When using MLE, the proposed single-timestamp scheme and one-way timestamp scheme exhibit a nearly identical performance, as the number of samples they observe is essentially the same. However, the proposed scheme further reduces communication overhead. Notably, in one-way message dissemination, the proposed NR-MLE algorithm significantly outperforms the conventional MLE. For instance, when *N* = 40, it improves clock skew and offset estimation accuracy by 2.98 times and 3.58 times, respectively. Moreover, one-way message dissemination using the NR-MLE algorithm even surpasses two-way message exchange using MLE in relation to estimation accuracy, despite the latter having more observation samples. Furthermore, the MSEs of both parameters asymptotically approach their respective CRLBs, validating the optimality of the NR-MLE algorithm. These results validate the fact that our approach significantly enhances time synchronization accuracy and efficiency in Industrial Internet of Things applications, while minimizing resource overhead.

Figure 5 and Figure 6 illustrate the MSEs of clock skew and the offset of estimation performance versus the increasing variance of random delay $\sigma^{2}$ when *N* = 40 in one-way message dissemination. It can be seen that the proposed NR-MLE algorithm outperforms MLE in both clock skew and offset estimation under identical random delay variance $\sigma^{2}$, especially when the variance is large. Furthermore, while increasing $\sigma^{2}$ degrades the performance of both approaches, MLE exhibits a significantly higher sensitivity. Specifically, when $\sigma^{2}$ increases from 0.1 to 1, MLE suffers severe performance deterioration, with clock skew and offset estimation accuracy declining by approximately 10.1 times and 10.2 times, respectively. In contrast, the NR-MLE algorithm shows only 5.7 times and 5.1 times performance degradation in skew and offset estimation, respectively, demonstrating the strong robustness of the proposed algorithm against increasing the variance of random delay.

Figure 7 shows the clock skew and offset estimation performance versus the increasing fixed delay *d* when the variances of random delay $\sigma^{2}$ = 1 and *N* = 40. Based on the previous analysis, the fixed delays vary depending on the distances between unsynchronized nodes and APs. Therefore, we change the fixed delay to simulate the performance of the proposed algorithm parameter estimation when the distance between nodes and APs changes. By adjusting the fixed delay in simulations, we observe that the proposed algorithm’s performance slightly degrades as the delay increases. However, it consistently outperforms the traditional MLE algorithm across all tested scenarios. Specifically, even when d = 1 ms, the proposed algorithm achieves a 1.97 times and 2.09 times higher accuracy in estimating clock skew and offset, respectively, compared to MLE. These results demonstrate the robustness of the proposed algorithm against increasing distances.

Figure 8 and Figure 9 show the effect of the regularization parameter λ on the clock skew and offset estimation performance when the simulated noise variance is 1 and N=40. It can be observed that the estimation performance of clock parameters remains relatively stable (with variations within 10–20%) for λ values ranging from 0.001 to 0.1. This insensitivity to λ demonstrates the robustness of the proposed NR-MLE algorithm. Note that the values of λ can correspond to various IIoT scenarios. Specifically, when λ is too small (e.g., <0.005), the algorithm prioritizes reconstruction fidelity, which may retain noise and degrade estimation accuracy under high-noise conditions. When λ is too large (e.g., >0.05), over-penalization risks oversmoothing genuine signal components, leading to biased estimates. Therefore, from the figures, we identified λ=0.01 as the optimal value that minimizes MSE for both clock skew and offset estimation. This choice effectively balances noise suppression and signal preservation.

Figure 10 shows the PDF of residual noise after the NR-MLE algorithm versus the original noise when the simulated noise variance is 1. It can be observed that the residual noise follows a Gaussian distribution with a mean of approximately zero (μ≈0). This indicates that the noise reduction process preserves the statistical properties of unbiased and symmetric fluctuations. Moreover, the variance of the residual noise ($\sigma^{2}=0.2655$) is 73% lower than that of the original noise ($\sigma^{2}=0.9998$), which is equivalent to a 3.7-fold reduction in dispersion. This significant suppression of noise amplitude aligns with the performance improvement observed in Figure 3 and Figure 4, demonstrating the effectiveness of the applied processing method.

Figure 11 and Figure 12 compare the mean absolute error (MAE) and 95% confidence intervals (CIs) between the proposed NR-MLE algorithm and conventional MLE for clock skew and offset estimation under identical noise conditions ($\sigma^{2}$ = 1). The results show that NR-MLE achieves significantly lower MAE values of 0.0068 (95% CI [0.0067, 0.0069]) for clock skew estimation and 0.2476 (95% CI [0.2439, 0.2512]) for offset estimation, representing 42.4% and 47.2% improvements over conventional MLE (MAE = 0.0118, 95% CI [0.0116, 0.0120] for skew; MAE = 0.4693, 95% CI [0.4623, 0.4762] for offset). A two-sample *t*-test confirms that these improvements are statistically significant (*p* < 0.01), validating the robustness of NR-MLE against random delay noise.

Figure 13 and Figure 14 show the MAE of clock skew and offset estimation between the proposed NR-MLE and conventional MLE algorithms with respect to the number of synchronization rounds *N* when the variance of random delay $\sigma^{2}$ = 1. It can be seen that when N=40, NR-MLE achieves 0.0064 skew MAE (1.71 times better than MLE’s 0.011) and 0.23ms offset MAE (1.95 times better than MLE’s 0.45ms). This performance not only meets but exceeds the <1 ms synchronization accuracy requirement specified in IEC/IEEE 60802 [33] for Industrial IoT applications, with the 0.23 ms offset error providing a 4.3 times safety margin for critical IIoT use cases including precision motor control, distributed sensor networks, and multi-robot coordination systems.

Figure 15 shows the convergence performance of the proposed NR-MLE algorithm when the number of synchronization rounds N=40 and the variance of random delay $\sigma^{2}$ = 1. The figure demonstrates a clear decreasing trend in error as the iteration count increases, eventually stabilizing, which indicates the robust convergence properties of the algorithm. Specifically, the initial error is relatively high but decreases rapidly during the first few iterations, suggesting that the algorithm efficiently approaches an optimal solution early on in the process. Beyond approximately 5 iterations, the rate of error reduction slows and the error stabilizes to a low level after around 10 iterations, confirming convergence. This behavior highlights the proposed algorithm’s computational efficiency, achieving significant optimization with relatively few iterations, as well as its numerical stability, maintaining minimal error fluctuations in later stages. Furthermore, the smooth, monotonic decline of the error curve—without noticeable oscillations or rebounds—validates the algorithm’s well-designed and robust nature.

Figure 16 shows the comparison of communication overhead between the timestamp scheme and the proposed single-timestamp scheme with increasing number of synchronization times. For the overhead analysis, similar to [21], we refer to the IEEE 802.15.4 standards where each timestamp occupies 4 bytes. It can be seen that as the number of synchronization rounds increases, the communication overhead of traditional one-way and two-way timestamp schemes increases linearly. However, the proposed single-timestamp scheme maintains a communication overhead of 4 bytes, which is significantly better than that of traditional schemes. This advantage is particularly prominent in large-scale distributed systems or high-frequency synchronization scenarios, providing a practical solution for reducing network load and improving system performance. As demonstrated by Pottie and Kaiser [34], transmitting 1 bit over 100 m consumes energy equivalent to executing 3 million instructions (3 Joules).

### 4.3. Computational Complexity Analysis

We compare the computational complexity of the proposed single-timestamp time synchronization scheme (with NR-MLE) to traditional synchronization schemes. Assuming the total number of synchronization rounds is *N*, the computational complexity of the one-way timestamp scheme with MLE is O(24N+3), while the two-way timestamp scheme with MLE increases to O(48N+30). In the proposed scheme, due to the small dimension of the timestamp matrix, the rank *p* of the decomposition matrix can be set to 1. Consequently, the computational complexity of the proposed single-timestamp time synchronization scheme with the NR-MLE algorithm is O((24+I)N+3), where *I* denotes the number of iterations in NR-MLE. As shown in Figure 8, the algorithm typically converges within 10 iterations, ensuring a low computational overhead. Thus, the proposed algorithm maintains comparable complexity to existing schemes, while achieving a significantly lower communication overhead and a higher estimation accuracy.

## 5. Conclusions

In this paper, a noise-robust-based MLE algorithm has been proposed to enhance the parameter estimation accuracy in time-sensitive IIoT applications, considering that the parameter estimation performance of MLE deteriorates by the random delay noise. The algorithm has adopted a two-phase approach. First, it systematically preprocesses the noisy timestamp matrix by decomposing and reconstructing its core components to isolate and mitigate random delay effects; then, it applies refined MLE estimation on the denoised data to achieve enhanced parameter estimation accuracy. Additionally, a single-timestamp time synchronization scheme has been proposed by utilizing the mechanism of AP periodically collecting sensor device data to reduce the bandwidth overhead and power consumption of time synchronization. The simulation results demonstrated that the proposed NR-MLE algorithm significantly improves estimation accuracy compared to the conventional MLE and exhibits a strong robustness against increasing delay variance. This work has provided a new perspective for parameter estimation in time synchronization systems. For future work, we plan to extend this framework to handle exponentially distributed delays, which may be caused by heavy channel contention or serious interference, and measure the synchronization delay under real interference in order to further improve the applicability to specific IIoT scenarios.

## Figures and Tables

**Figure 1 entropy-27-00927-f001:**
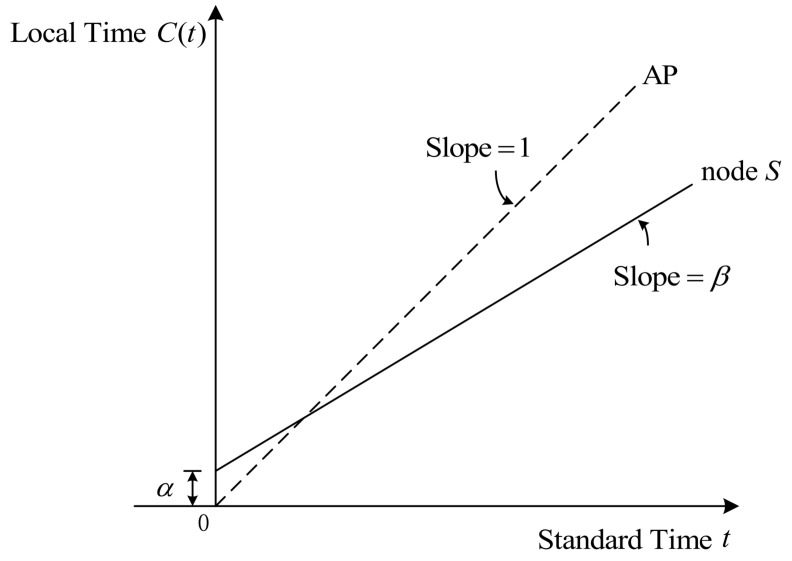
Clock model.

**Figure 3 entropy-27-00927-f003:**
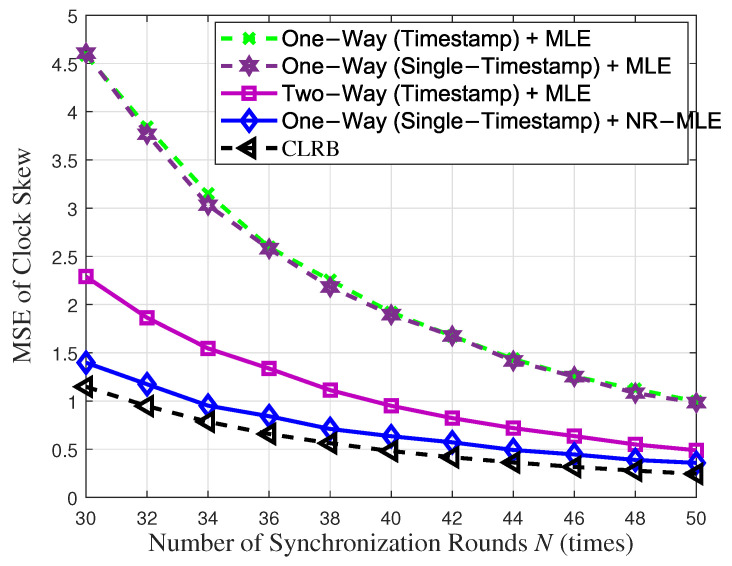
Clock skew estimation performance with respect to synchronization rounds *N*.

**Figure 4 entropy-27-00927-f004:**
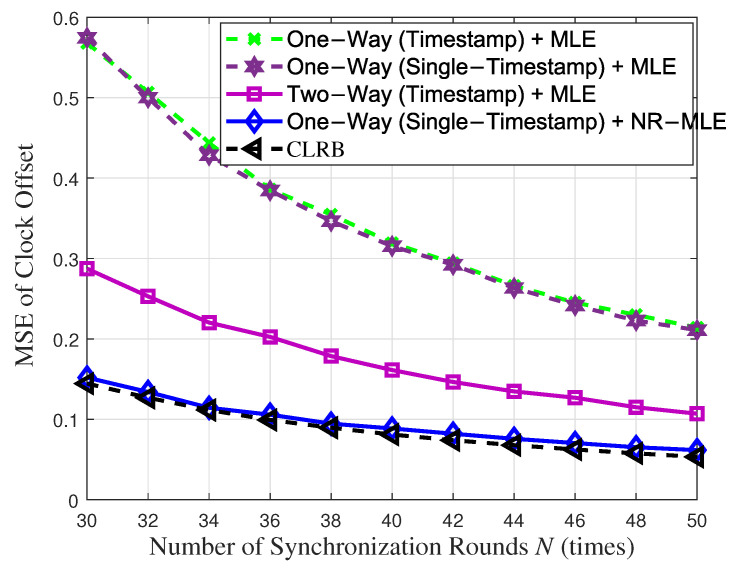
Clock offset estimation performance with respect to synchronization rounds *N*.

**Figure 5 entropy-27-00927-f005:**
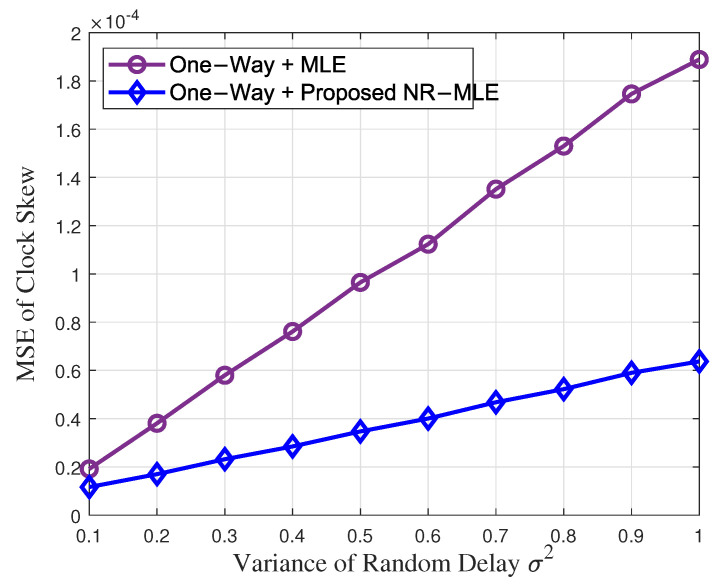
Clock skew estimation performance versus the increasing variance of random delay $\sigma^{2}$.

**Figure 6 entropy-27-00927-f006:**
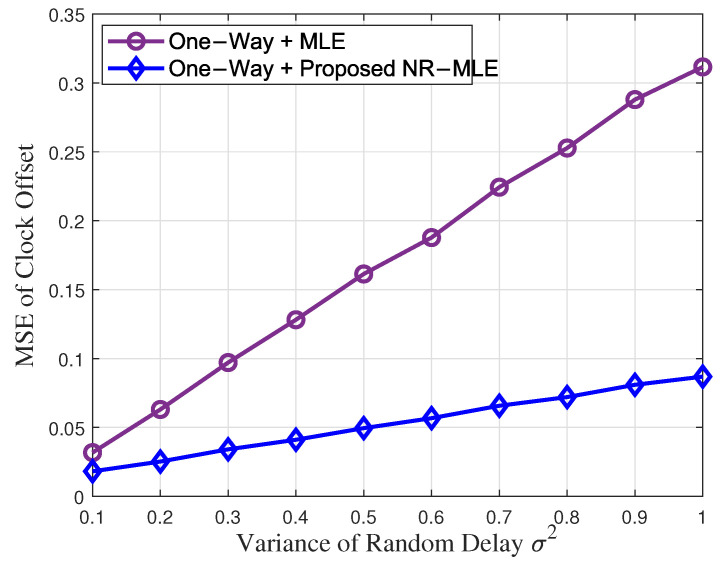
Clock offset estimation performance versus the increasing variance of random delay $\sigma^{2}$.

**Figure 7 entropy-27-00927-f007:**
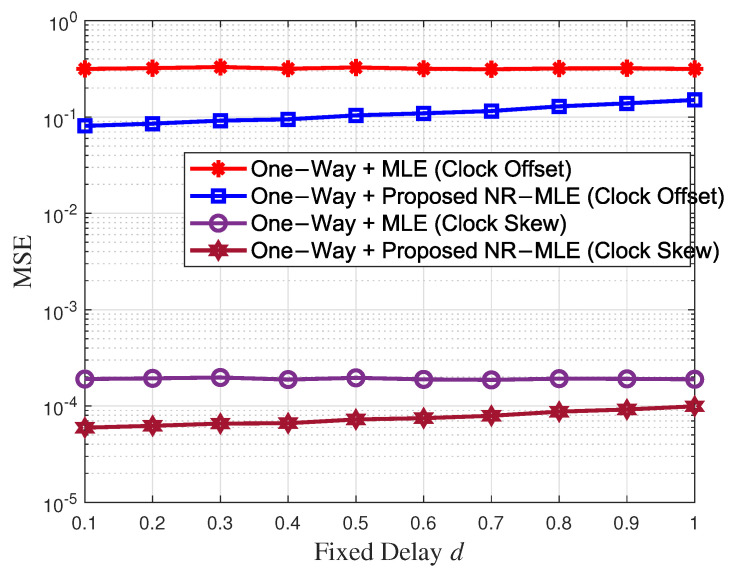
Clock skew and offset estimation performance versus the increasing fixed delay *d*.

**Figure 8 entropy-27-00927-f008:**
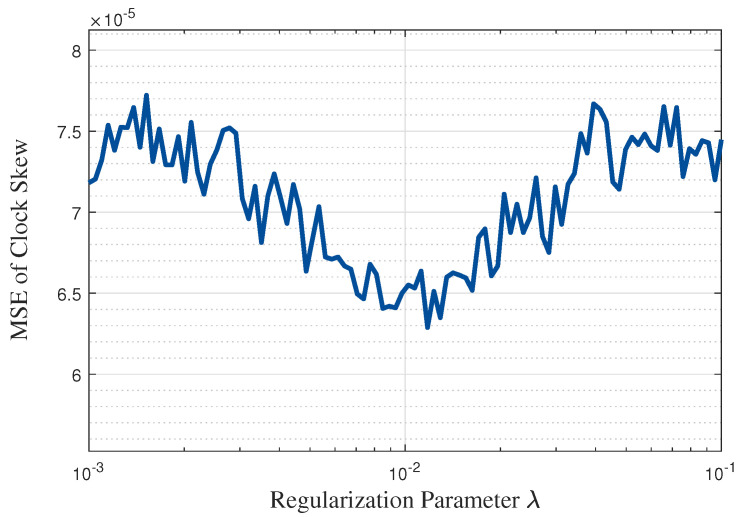
The effect of the regularization parameter λ on the clock skew estimation performance.

**Figure 9 entropy-27-00927-f009:**
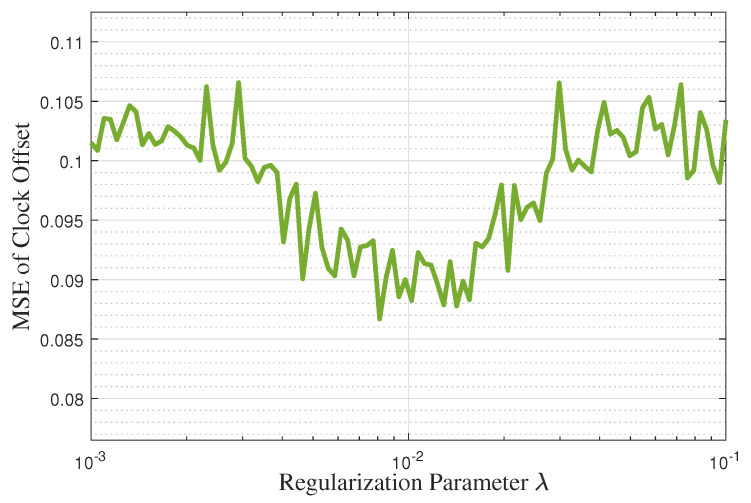
The effect of the regularization parameter λ on the clock offset estimation performance.

**Figure 10 entropy-27-00927-f010:**
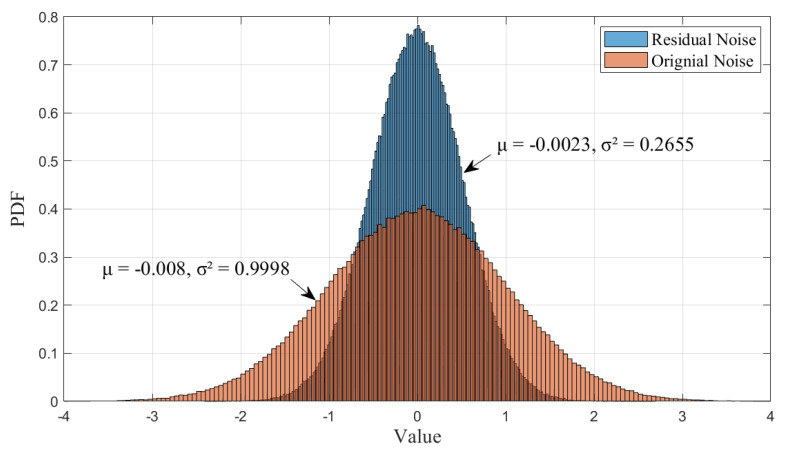
PDF of residual noise and original noise when simulated noise variance is 1.

**Figure 11 entropy-27-00927-f011:**
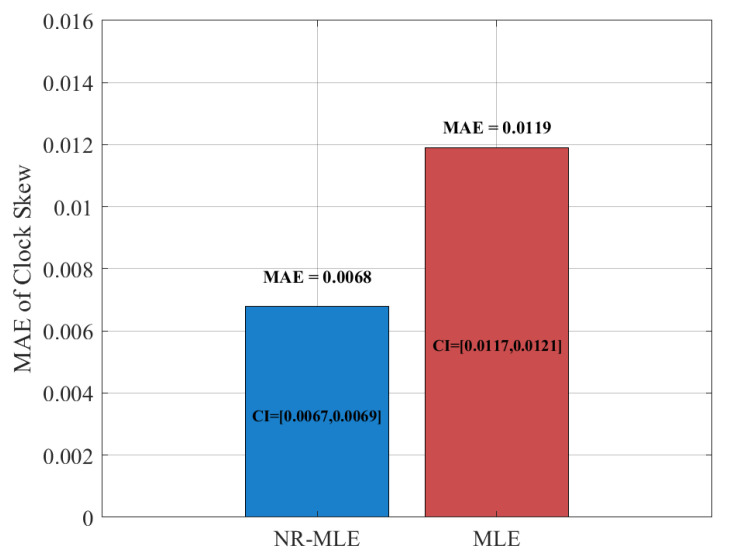
MAE and CIs between the proposed NR-MLE algorithm and conventional MLE for clock skew estimation.

**Figure 12 entropy-27-00927-f012:**
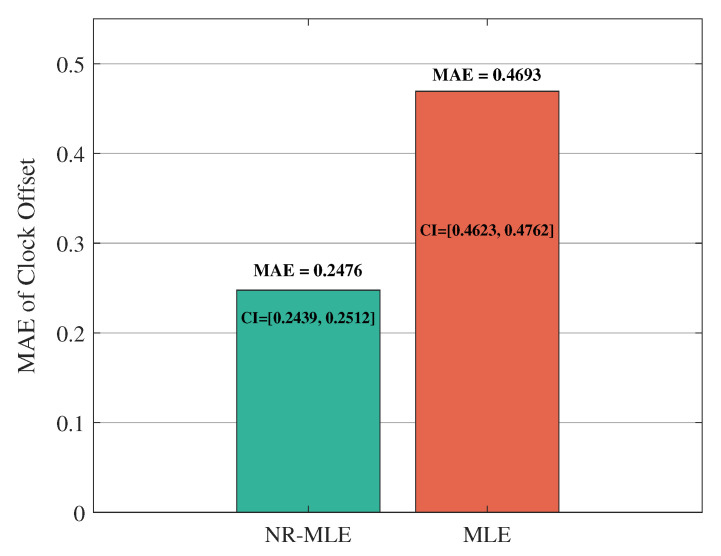
MAE and CIs between the proposed NR-MLE algorithm and conventional MLE for clock offset estimation.

**Figure 13 entropy-27-00927-f013:**
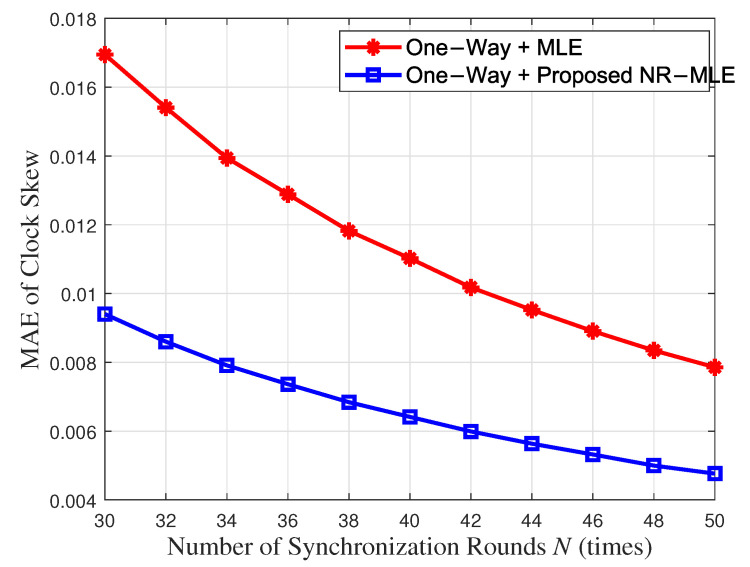
MAE of clock skew with respect to synchronization rounds *N*.

**Figure 14 entropy-27-00927-f014:**
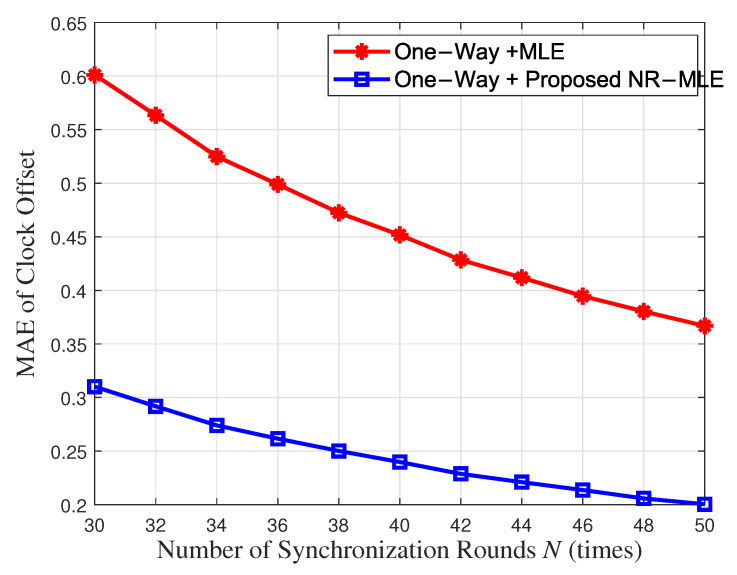
MAE of clock offset with respect to synchronization rounds *N*.

**Figure 15 entropy-27-00927-f015:**
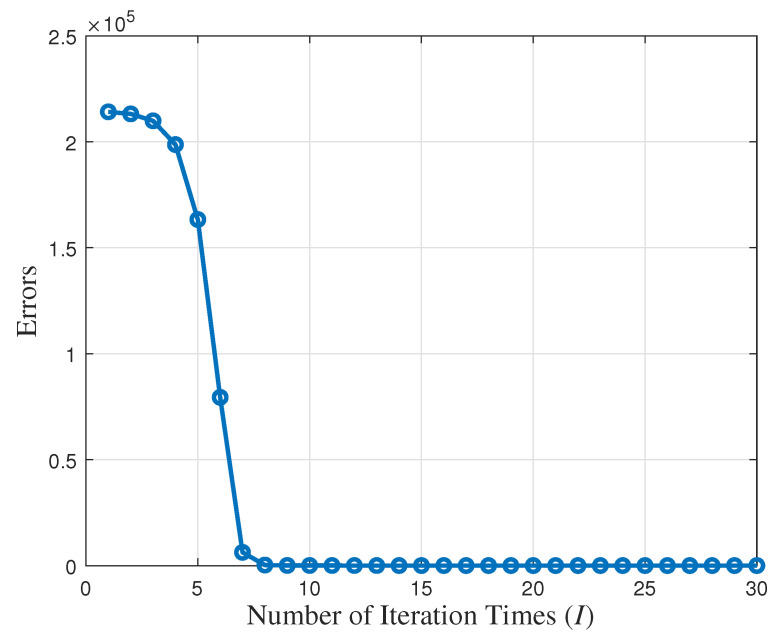
Convergence performance of the proposed NR-MLE algorithm.

**Figure 16 entropy-27-00927-f016:**
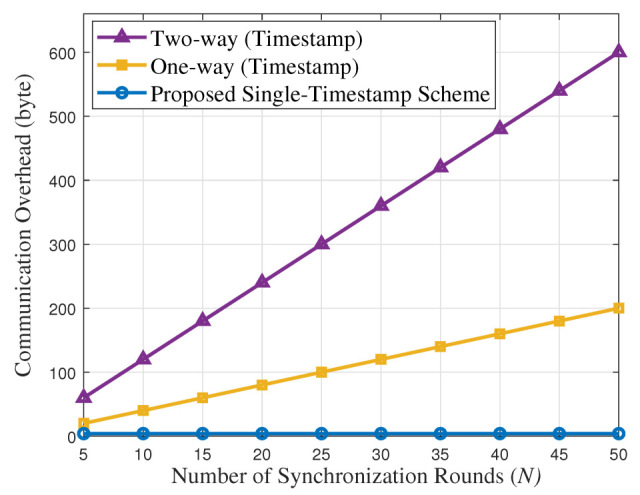
Comparison of communication overhead.

**Table 1 entropy-27-00927-t001:** Simulation parameters.

Notation	Description	Value
β	Relative clock skew	[0.99, 1.01]
α	Relative clock offset	[−0.02, 0.02] ms
*d*	Fixed delay	[0, 0.2] ms
$\sigma^{2}$	Variance of random delay	[0.1, 1]
*p*	Target rank	1
λ	Regularization coefficient	[0.001, 0.1]
*N*	Synchronization rounds	30:5:50
δ	Fixed interval	1 ms
max	Simulation times	10,000

## Data Availability

The data presented in this study are available on request from the corresponding author. The data are not publicly available due to copyright.

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
