# Peer review of "Noise-Robust-Based Clock Parameter Estimation and Low-Overhead Time Synchronization in Time-Sensitive Industrial Internet of Things"

_entropy, 2025, doi:10.3390/e27090927_

Round 1

Reviewer 1 Report

Comments and Suggestions for Authors

This paper works on a timely research topic for time-sensitive IoT, i.e., time synchronization. This paper proposes a noise-robust MLE to improve the clock parameter estimations, together with CRLB calculation. A gradient descent method is employed to further reduce the noise. Based on the estimation, a single-timestamp synchronization scheme is proposed. Overall, this paper is well prepared, and the results show effectiveness. The following comments can help to further improve this paper.

  1. The clock model is not well defined in Fig.1. In the figure, t, C(t) are not described.
  2. In the delay analysis---Section 2.2, the portions of the delays are not analyzed. Are those delays exactly the same? How to address them?
  3. Equations (4)-(6) can be presented in a single line, instead of three standalone lines.
  4. In (8), please explain the dimensions, e.g., R^NX2, T^SR.
  5. In the energy-efficient synchronization research, there thrive many new methods which are not well discussed in the introduction, e.g., a one-way time synchronization scheme for practical energy-efficient LoRa networks, a microsecond energy-efficient LoRa time synchronization based on low-layer timestamping.
  6. The optimization objectives require further explanation.
  7. Please detailedly describe the simulation platform. Note that, the time synchronization accuracy is highly affected by the simulation setup, e.g., delay generation, clock time simulation, etc.
  8. The presentation of the PDF is good. Could you please provide more PDF statistics figures for your evaluation results?
  9. By the way, can you simulate a longer distance to show the robustness of the proposed clock parameter estimations?

This paper can be considered for acceptance after a thorough revision.

Reviewer 2 Report

Comments and Suggestions for Authors

1. How sensitive is the proposed NR-MLE algorithm’s estimation accuracy to the selection of the regularization parameter λ and the target rank p? Has the algorithm been extensively evaluated across a wide range of these hyperparameters, and how does this affect its robustness in various IIoT scenarios?

  1. 2. ​​The NR-MLE algorithm's core innovation—using matrix decomposition and gradient descent for noise reduction—is well-formulated theoretically, and Figure 8 shows promising convergence within ~10 iterations. However, the simulation relies solely on synthetic Gaussian noise  which may not reflect real IIoT environments where delays often exhibit non-Gaussian characteristics due to channel contention or interference. It is recommended to expand the non-Gaussian noise simulation and measure the synchronization delay under real interference.
  2. The simulation results convincingly show the advantage of NR-MLE over conventional MLE, especially in high-noise regimes. Nonetheless, the manuscript would benefit from a more detailed statistical analysis, such as confidence intervals or hypothesis testing, to quantify the significance of the observed improvements. Additionally, reporting absolute synchronization error, rather than just MSE, would facilitate practical interpretation and comparison with industrial synchronization requirements.

Round 2

Reviewer 1 Report

Comments and Suggestions for Authors

Thank you for the revision. My previous comments have been addressed. This paper is suggested for acceptance.

Reviewer 2 Report

Comments and Suggestions for Authors

1、 The authors have addressed the sensitivity of the regularization parameter λ and the target rank p in Figures 8 and 9. The results demonstrate robustness across a reasonable range of values. However, a brief discussion on how these parameters may be optimally selected in practical IIoT scenarios would enhance the manuscript’s applicability.

2、The inclusion of additional noise models, as recommended, has strengthened the manuscript. The simulation results under non-Gaussian noise provide valuable insights into the algorithm’s robustness. If feasible, a brief mention of potential challenges in extending the algorithm to real-world IIoT interference scenarios would add further clarity.

3、The addition of confidence intervals and hypothesis testing effectively quantifies the significance of the improvements. The inclusion of absolute synchronization error metrics satisfies practical concerns, aligning well with industrial requirements. This addition is highly commendable.

  •